# Cortical Correlates of Increased Postural Task Difficulty in Young Adults: A Combined Pupillometry and EEG Study

**DOI:** 10.3390/s22155594

**Published:** 2022-07-26

**Authors:** Melike Kahya, Ke Liao, Kathleen M. Gustafson, Abiodun E. Akinwuntan, Brad Manor, Hannes Devos

**Affiliations:** 1Hinda and Arthur Marcus Institute for Aging Research, Hebrew SeniorLife, Harvard Medical School, 1200 Centre St., Boston, MA 02131, USA; bradmanor@hsl.harvard.edu; 2Hoglund Biomedical Imaging Center, University of Kansas Medical School, Kansas City, KS 66160, USA; kliao@kumc.edu (K.L.); kgustafson@kumc.edu (K.M.G.); 3Department of Neurology, University of Kansas Medical Center, Kansas City, KS 66160, USA; 4Department of Physical Therapy, Rehabilitation Science, and Athletic Training, University of Kansas Medical Center, Kansas City, KS 66160, USA; aakinwuntan@kumc.edu (A.E.A.); hdevos@kumc.edu (H.D.)

**Keywords:** pupillary response, electroencephalogram, dual-task, postural control, young adults

## Abstract

**Highlights:**

**Abstract:**

The pupillary response reflects mental effort (or cognitive workload) during cognitive and/or motor tasks including standing postural control. EEG has been shown to be a non-invasive measure to assess the cortical involvement of postural control. The purpose of this study was to understand the effect of increasing postural task difficulty on the pupillary response and EEG outcomes and their relationship in young adults. Fifteen adults completed multiple trials of standing: eyes open, eyes open while performing a dual-task (auditory two-back), eyes occluded, and eyes occluded with a dual-task. Participants stood on a force plate and wore an eye tracker and 256-channel EEG cap during the conditions. The power spectrum was analyzed for absolute theta (4–7 Hz), alpha (8–13 Hz), and beta (13–30 Hz) frequency bands. Increased postural task difficulty was associated with greater pupillary response (*p* < 0.001) and increased posterior region alpha power (*p* = 0.001) and fronto-central region theta/beta power ratio (*p* = 0.01). Greater pupillary response correlated with lower posterior EEG alpha power during eyes-occluded standing with (*r* = −0.67, *p* = 0.01) and without (*r* = −0.69, *p* = 0.01) dual-task. A greater pupillary response was associated with lower CoP displacement in the anterior–posterior direction during dual-task eyes-occluded standing (*r* = −0.60, *p* = 0.04). The pupillary response and EEG alpha power appear to capture similar cortical processes that are increasingly utilized during progressively more challenging postural task conditions. As the pupillary response also correlated with task performance, this measurement may serve as a valuable stand-alone or adjunct tool to understand the underlying neurophysiological mechanisms of postural control.

## 1. Introduction

Postural control enabling a bipedal stance is essential to a variety of human goal-directed movements. Although early animal studies suggested limited participation of the cerebral cortex in controlling posture, the cerebral cortex plays a major role in standing postural control in humans. This is particularly the case when an individual must regulate their posture while completing concurrent cognitive tasks and/or when visual feedback is reduced, altered, or occluded [1,2]. It is important to understand the cortical correlates of increased postural task difficulty associated with visual occlusion and dual-tasking to better assess fall risk and to develop precision medicine approaches for individualized rehabilitation interventions.

The pupillary response is a non-intrusive, real-time neurophysiological measure of cognitive workload (or mental effort). An increased pupillary response is believed to reflect greater cognitive workload during completion of a challenging cognitive and/or motor task [3]. Moreover, there is growing evidence that the pupillary response is a reliable and valid neurophysiological measure of brain activity during challenging postural control tasks [4]. Studies demonstrated that the pupillary response significantly increased from single standing to dual-task standing and from eyes-open to eyes-occluded conditions, in older adults with and without Parkinson’s disease [5]. A study from our laboratory investigated the effect of dual-task standing on pupillary response but the additional cognitive task (auditory Stroop) was not challenging enough to elicit an increased pupillary response [3]. The present study used a more challenging cognitive task (two-back test) [6] to understand the effect of an additional cognitive task during standing in healthy young adults.

The physiologic mechanism of the pupillary response originates from the activation of the locus coeruleus [7]. The locus coeruleus is responsible for the regulation of physiological arousal and attention [8]. When the locus coeruleus is activated, it sends inhibitory signals to the Edinger–Westphal nucleus, which is part of the parasympathetic nervous system. The Edinger–Westphal nucleus then inhibits the sphincter pupillae muscle, which leads to increased pupil dilation [9]. Therefore, the activation of the locus coeruleus is associated with an increased pupillary response due to increased cognitive workload [10]. Although the pupillary response recording has no spatial resolution, the temporal resolution ranges from 1 Hz to 1000 Hz [11], comparable to electroencephalography (EEG) frequencies. In the advent of mobile imaging, pupillary response may become a preferred neurophysiological measure of cognitive workload during balance and gait studies, since measuring the pupillary response is more cost-effective, less intrusive, and requires less preparation and processing time compared to EEG.

At the same time, EEG is an established non-invasive measure to assess the cortical involvement of postural control [12,13]. The strengths of EEG are its portability and high temporal resolution, which allow the quantification of cortical involvement in real-time during standing [13]. However, EEG has the limitations of low spatial resolution and susceptibility to movement artifacts [14]. Advances in EEG signal processing techniques such as independent component analysis enable investigators to separate brain activity from movement artifacts and to enhance the spatial resolution [15].

EEG frequency bands are direct measurements of brain activity [16]. Alpha power (8–13 Hz) has been shown to decrease with increased task difficulty during upright stance posture in young adults [17,18]. While a widespread decrease in absolute alpha power is supposed to reflect global processes of increased attention and alertness [19,20], activity within the posterior channels appears to be associated with sensory and movement-related information processing [17]. It is thus possible that alpha power is not only linked to attention or alertness, but also to sensory and movement-related information processing, especially in the context of standing postural control. Indeed, an inverse relationship between alpha power activity and the pupillary response was shown while individuals were engaging in a cognitive task [21,22,23], suggesting that these outcomes may reflect similar underlying neurophysiologic processes important to postural control. However, no previous studies investigated this inverse relationship between the pupillary response and alpha power during postural control in healthy young adults.

In addition, the power ratio of EEG slow-wave (theta) to fast-wave (beta) activity has been shown to reflect cognitive activity related to attention and working memory processes. The theta/beta ratio has shown to be a robust measure of cognitive processing and strongly correlates with EEG event-related potential (ERP) P3 latency [24]. The P3, a positive peak that appears with a latency between 250 and 500 ms in the ERP, has been implicated in attention and working memory processes [25]. Currently, it remains unknown whether the EEG theta/beta ratio shows an association with increased postural task difficulty in young adults.

The aim of this study was to understand the effect of increasing postural task difficulty on the pupillary response and EEG outcomes and their relationship in young adults. We hypothesized that both the pupillary response and EEG outcomes would reflect increased postural task difficulty. We also investigated the relationship between the pupillary response and EEG outcomes during increased postural task difficulty in healthy young adults. We hypothesized that there would be an inverse relationship between the pupillary response and EEG alpha power in various postural demanding conditions in healthy young adults. Demonstrating the validity of the pupillary response against a standard measure of brain activity during increased postural task difficulty might lead to increased usage of the pupillary response to capture the underlying neurophysiological mechanisms of increased postural task difficulty in healthy and disease populations. A better understanding of the neurophysiological signatures of postural control may inform more adequate assessment and treatment strategies to mitigate balance impairments and falls.

## 2. Methods

### 2.1. Participants

Fifteen young adults were tested at the University of Kansas Medical Center, Hoglund Biomedical Imaging Center. Eligible participants were between 18 and 30 years old, with self-reported normal hearing and vision, were able to ambulate independently, with intact cognition measured by Montreal Cognitive Assessment (scored > 26), and were able to speak and understand the English language. Exclusion criteria included a history of neurological or vestibular conditions, current visual field loss or reduced visual acuity that could not be resolved by corrective lenses or glasses, and musculoskeletal problems that might affect standing and balance conditions. None of the recruited participants were excluded based on these criteria. This study was approved by the Human Subjects Committee Institutional Review Board of the University of Kansas Medical Center. All methods were performed in accordance with the Declaration of Helsinki. Participants provided written consent.

### 2.2. Protocol

Demographic data including age, sex, education level, and handedness were collected. The Montreal Cognitive Assessment (MoCA) [26] was administered to measure global cognitive function. Participants were then fitted with Tobii Pro 2 eye tracking glasses (Tobii Tech., Stockhom, Sweden) to measure the pupillary response during subsequent testing of standing posture. In addition, participants were fitted with a 256-channel high-density EEG net (Magstim EGI, Eugene, OR, USA). Participants were tested in a room with no windows. The temperature and ambient lighting were kept constant across conditions. Participants were asked to step on a force plate (AMTI OPT464508-1000, Advanced Mechanical Technology, Inc., Watertown, MA, USA) that recorded Center of Pressure (CoP) displacement in the anterior–posterior (AP) and medio–lateral (ML) directions with a sampling frequency of 100 Hz. Participants were instructed to stand on the force plate while wearing their shoes and placing their feet oriented at 14° with heel centers 17 cm apart (Figure 1). An auditory 2-back task was presented using E-Prime^®^ (version 3.0, Psychology Software Tools, Sharpsburg, PA, USA). Behavioral responses (button-press) were recorded with Net Station software (Magstim EGI, Eugene, OR, USA). As assessments took around 2 h, we mitigated the effect of fatigue by giving breaks between conditions and rest periods anytime during the study. To ensure subject safety, participants wore a gait belt and a research assistant stood behind the participants to prevent falls. Participants were asked to complete four different conditions in a randomized order. The standing conditions took approximately 20 min. In order to minimize the participants’ fatigue from standing and reduce sleepiness, the EEG preparation was done when the participant was sitting. Participants were given breaks between standing tasks and allowed to sit to rest.

Standing with eyes open (control condition): Participants stood on a force plate and maintained an upright standing posture for 320 s.Dual-task eyes open: Participants stood on the force plate for 320 s while concurrently completing an auditory 2-back test by using a thumb-press button.Standing with eyes occluded: Participants stood on a force plate for 320 s with a sleep mask covering the eye tracker. We asked our participants to keep their eyes open to record the pupillary response.Dual-task eyes occluded: Participants stood on a force plate for 320 s while completing the 2-back test with their eyes occluded.

**2-Back Test** [6]: During dual-task eyes-open and eyes-occluded conditions, individuals performed the auditory 2-back test. In the 2-back test, participants heard a series of letters delivered through a speaker. They were instructed to press a hand-held thumb-press button when the current letter was the same as the letter that they heard 2 positions back. The auditory stimuli and button-press were synchronized with the EEG system. Each test comprised 160 trials, including 50 trials that needed a response (target, 31.25%) and 110 trials for which a response was not required (non-target, 68.75%). Each letter was presented in random order for 1500 ms from the speaker, followed by a blank interstimulus interval for 2000 ms, with a random jitter of ±100 ms. Each task time was ∼5 min. The number of correct presses and response times were analyzed in MATLAB.

### 2.3. Outcome Variables

#### 2.3.1. Pupillary Response

Pupil diameter was extracted at 100 Hz from EyeWorks Analyze software. When solely measuring the change in the raw pupil size, there are potential limitations, such as the light reflex and movement artifacts interfering with the pupil size. We minimized these potential confounders by keeping the lighting in the room constant and having participants focus on a picture of dots on the wall during eyes-open conditions. In addition, we used the Index of Cognitive Activity (ICA) algorithm, calculated through the EyeWorks Analyze software, to differentiate the pupillary response due to workload from the light reflex [27]. This algorithm computes the number of unusual increments in pupil size at 1 Hz. These values are then transformed into a continuous scale ranging between 0 (no cognitive workload) and 1 (maximum cognitive workload). Based on this algorithm, the noisy signals are reduced to nearly zero [27]. The mean ICA was calculated after each condition for all groups.

#### 2.3.2. EEG

Continuous EEG was digitized at 1000 Hz. Data were online referenced to Cz and offline filtered from 0.50 to 30 Hz using EGI software. All other EEG processing was done in EEGLab [28]. Only 183 channels were used in the analyses to decrease processing complexity by removing the peripheral EEG channels, such as those on cheeks. In addition, various artifacts unrelated to cognitive functions, including ocular movement or cardiovascular signals, were identified and removed using independent component analysis. Continuous EEG data were segmented into each trial with a length of 100 ms for baseline period pre-stimulus and 1500 ms after stimulus onset. Epoched data rejection methods in EEGLAB were used. Specifically, outlier values were detected by finding abnormal values with amplitude larger than 150 µV or lower than −150 µV. Maximal absolute slope of the linear trend of epochs was set as 100 µV /epoch. Minimal linear regression R-square value was set as 0.3. Peaky distribution of activity was detected by setting the kurtosis limit as 7 standard deviations for both single and global channels. Abnormal spectra thresholds were set as +/−100 dB by using Slepian multitaper for spectrum computation. On average, 26.5 epochs (16.56%) were removed in artifact removal, with 14.57% in the dual-task eyes-occluded condition and 18.56% in the dual-task eyes-open condition. Signals from bad electrodes were interpolated using surrounding electrode data. To calculate the theta (4–7 Hz), alpha (8–13 Hz), and beta (13–30 Hz) frequency band power for all four conditions, EEGlab’s Spectopo function was used to extract power spectral density from all electrodes, which uses Welch’s method on the 1 s epochs with 50% overlap between its calculation windows. One participant was removed from the analyses due to not being able to collect EEG data due to a technical problem. Another participant was removed due to not being able to complete all tasks due to feeling light-headed after completing half of the tasks. The absolute mean power of all electrodes was calculated across all subjects for each condition. Based on the results of the scalp maps, we further investigated the fronto-central region (AF7, AF8, AF3, AF4, C3, F3, C1, P3, P1, CP2, C4, C6, FC6, F6, F2, Fz, FCz, Cz) theta/beta power ratio and posterior region (POz, P7, P5, Pz, PO3, P9, O1, Oz, PO4, O2, P4, P6, P10, CP6, TP8) alpha power.

#### 2.3.3. COP

The force platform collected forces Fx, Fy, and Fz and movements Mz, My, and Mz. CoP displacement was calculated in the x and y directions with the following calculations:CoP_x_ = −My/Fz
CoP_Y_ = −Mx/Fz

A custom MATLAB code (MathWorks, Natick, MA, United States) employed a 4th-order Butterworth low-pass filter with a cut-off frequency of 10 Hz and resampled the data at 100 Hz. The CoP displacement in the AP and ML directions was calculated by using NetForce Ver. 3.5.3 software for each condition.

## 3. Statistical Analysis

Demographic and clinical variables were reported as mean and standard deviation. We used Kolmogorov–Smirnov and Levene’s tests to test for normality and homogeneity of variance in the pupillary response, posterior EEG alpha power, fronto-central EEG theta/beta ratio, and CoP displacement. Repeated-measures ANOVA and LSD post-hoc tests were employed to calculate the within-group change in the pupillary response, posterior EEG alpha power, fronto-central EEG theta/beta ratio, and CoP displacement across the four conditions. Pearson’s correlation was used to analyze the relationship between pupillary response and EEG outcomes and CoP displacement. The results were interpreted as follows: >0.70 is strong, 0.50–0.70 is moderate, 0.30–0.50 is weak [29]. All statistical analyses were performed with IBM SPSS Statistics v.26 software (IBM, Armonk, NY, USA). *p*-values < 0.05 were considered statistically significant. 

## 4. Results

Participants (*n* = 15) were on average 25.6 ± 2.9 years old and scored 28.8 ± 1.5 on the MoCA scale. The average years of education was 18.6 ± 1.6 for the participants (Table 1). 

Figure 2 displays the effect of increased postural task difficulty on the pupillary response. The response increased significantly as a result of increasing task difficulty during postural control (*p* < 0.001). Post-hoc analysis demonstrated that the pupillary response increased from standing with eyes open (control condition) to dual-task eyes open (*p* = 0.01), standing with eyes occluded (*p* < 0.001), and dual-task eyes occluded (*p* < 0.001). In addition, the pupillary response increased from standing with eyes occluded to dual-task eyes occluded (*p* < 0.01). Lastly, a significant difference was present between dual-task eyes open and dual-task eyes occluded (*p* < 0.01).

Similarly, posterior alpha power showed a significant condition effect (*p* = 0.001). Post-hoc analysis demonstrated that posterior alpha power significantly increased from standing with eyes open (control condition) to standing with eyes occluded (*p* = 0.005) and dual-task eyes occluded (*p* = 0.009). Moreover, there was a significant increase from dual-task eyes open to dual-task eyes occluded (*p* = 0.007) (Figure 3).

In addition, increasing task difficulty resulted in changes in the fronto-central EEG theta/beta ratio (*p* = 0.01). Post-hoc analysis revealed that the fronto-central theta/beta ratio significantly increased from standing with eyes open to dual-task eyes open (*p* = 0.008) and to dual-task eyes occluded (*p* = 0.02) (Figure 4).

A visual depiction of the alpha power and theta/beta power ratios across standing conditions is presented in Figure 5. Scalp maps demonstrated increased posterior alpha power during the eyes-occluded conditions compared to eyes-open conditions, whereas the fronto-central theta/beta power ratio had increased activation during dual-task conditions compared to single standing.

A significant negative correlation was observed between the pupillary response and posterior EEG alpha power during the standing with eyes occluded condition (*r* = −0.69, *p* = 0.01) (Figure 6). In addition, there was a significant negative correlation between the pupillary response and posterior EEG alpha power during the dual-task eyes-occluded condition (*r* = −0.67, *p* = 0.01) (Figure 6). There were no significant correlations between the pupillary response and posterior EEG alpha power during standing with eyes open and dual-task eyes-open conditions. Moreover, there were no significant correlations between the pupillary response and fronto-central EEG theta/beta ratio during the conditions.

No significant change was observed with increased postural task difficulty in the CoP displacement in the AP (*p* = 0.82) and ML directions (*p* = 0.17). Moreover, the auditory two-back test results showed that, on average, participants responded accurately 95.4 ± 4.28% of the time to the target and non-target stimuli. The average response time was 736.4 ± 107.4 ms. A correlation analysis between the pupillary response and CoP displacement in the AP direction demonstrated a moderate association during dual-task eyes occluded (*r* = −0.60, *p* = 0.04). Moreover, a moderate correlation was observed between posterior EEG alpha power and CoP displacement in the AP direction during dual-task eyes occluded (*r* = 0.61, *p* = 0.03).

## 5. Discussion

The main goal of this study was to understand the effect of increasing postural task difficulty on the pupillary response and EEG outcomes and their relationship in young adults. Our results showed that the pupillary response was sensitive to the effect of increased postural task difficulty during both dual-tasking and visual occlusion, whereas posterior EEG alpha power was only sensitive to visual occlusion, and the fronto-central EEG theta/beta power ratio was only sensitive to dual-tasking. Moreover, the results showed a strong inverse correlation between the pupillary response and posterior EEG alpha power during standing with eyes occluded and dual-task eyes-occluded conditions. Previous studies demonstrated an increased pupillary response and decreased EEG alpha power during cognitive testing, suggesting that these neurophysiological outcomes reflect increased cognitive workload and attention [21,22]. In the present study, we measured these outcomes during increased postural task difficulty in young adults and observed similar results. These findings suggest that the pupillary response and EEG alpha power appear to capture the same cortical processes that are increasingly utilized when standing under difficult conditions.

The pupillary response, posterior EEG alpha power, and fronto-central theta/beta power ratio all increased with increased postural task difficulty. Previous studies have shown that the pupillary response increases significantly from a single task to dual-task balance conditions in healthy young and older adults [3,5]. Furthermore, global alpha power has been implicated in attentional aspects of balance control, whereas posterior alpha activity indicates task-specific movement and sensory information processing [17,30]. Several studies demonstrated that increased balance difficulty is associated with decreased global alpha activity while the eyes remain open [17,18,31]. In our study, we increased the postural task difficulty by dual-tasking and occluding the eyes. It is known that alpha power is profoundly involved in sensory processing and eye closure leads to increased alpha power, especially in the occipital lobe [32]. Therefore, the results of the present study suggest that increased alpha power within the posterior region was related to increased sensory information processing while participants were trying to maintain their balance during eyes-occluded conditions. Future studies should consider controlling eyes-occluded conditions while measuring the change in alpha activity during increased postural task difficulty.

In addition, previous studies demonstrated increased EEG theta power with increased task difficulty during postural control in young adults. Theta power is associated with attention, executive function, and working memory and has been shown to correlate with motor task difficulty in several populations [33,34]. Peterson et al. demonstrated that individuals who were standing while engaging in a cognitive task demonstrated higher theta power compared to the single standing position [35]. Similar results were observed in other studies that showed that EEG theta power increased with dual-tasking, standing on foam, or visual occlusion [36,37,38,39]. In addition, beta activity suppression was observed during gait control and challenging balance tasks [35,40,41]. Beta power in the parietal and central cortical regions has been shown to decrease with challenging gait and balance conditions, indicating that beta power is associated with motor inhibition [40]. It is possible that increased theta and decreased beta power are both involved during active postural control and may fluctuate as task difficulty changes. One way to better understand these fluctuations is to investigate the theta/beta ratio during challenging balance tasks. Our results extend previous findings that young adults had an increased theta/beta power ratio especially from single standing to dual-task standing conditions.

Although we found a significant change in neurophysiological responses, there were no changes with increased postural task difficulty in CoP displacement in the AP and ML directions. It is possible that the healthy young adults were successful in compensating for increased task difficulty by showing higher brain activity measured by the pupillary response and EEG, but due to the ceiling effect, the force platform could not record it. Therefore, neurophysiological tools might be more sensitive than the biomechanical outcomes to capture subtle changes in balance performance. Biomechanical measures are standard clinical measures in rehabilitation practice to understand balance performance [42]. Neurophysiological tools could potentially be used to improve rehabilitation outcomes by detecting subtle changes in balance, thus improving our understanding of postural control.

This study has some limitations. EEG recording is highly susceptible to various forms and sources of noise, which makes the signal-to-noise ratio one of the methodological challenges in analyzing EEG data. Although we recruited healthy young adults without any balance deficits, motion artifacts such as postural sway might have affected the data. In this study, we utilized strict methods to exclude EEG artifacts during postural control. To illustrate, individuals were asked to look straight ahead to reduce eye and neck movements during standing. In addition, independent component analysis was used to remove blinks and high-frequency artifacts associated with muscle tone. Lastly, we carefully visualized data and removed any other artifacts, such as heartrate artifacts, from the dataset. We believe that these carefully chosen methods ensured an EEG signal that represented actual brain activity during postural control. Secondly, there was no hardware synchronization between the pupillary response and EEG data during recording. We manually started and stopped the data collection at the same time on both devices and used time-locked triggers to indicate events (when letters were played during the auditory two-back test). However, this limited our ability to provide an image regarding moment-to-moment pupillary response in correlation with EEG outcomes. Lastly, we averaged frequency bands for a subset of channels that represent the fronto-central and posterior regions. It is known that EEG has low spatial resolution and, to maintain balance, all the brain lobes (frontal, temporal, parietal, and occipital) are needed to work synchronously. Therefore, limiting our selection criteria to only several regions might not represent all the brain activity that is required for postural control.

## 6. Conclusions

The pupillary response demonstrated a greater increase with increased postural task difficulty during both dual-tasking and visual occlusion, whereas posterior EEG alpha power was only sensitive to visual occlusion and the fronto-central EEG theta/beta power ratio was only sensitive to dual-tasking in young adults. In addition, a greater pupillary response correlated with lower posterior EEG alpha power during standing eyes-occluded and dual-task eyes-occluded conditions. In the future, the pupillary response could be a potential tool to understand the cortical correlates of increased postural task difficulty in healthy older adults and disease populations.

## Figures and Tables

**Figure 1 sensors-22-05594-f001:**
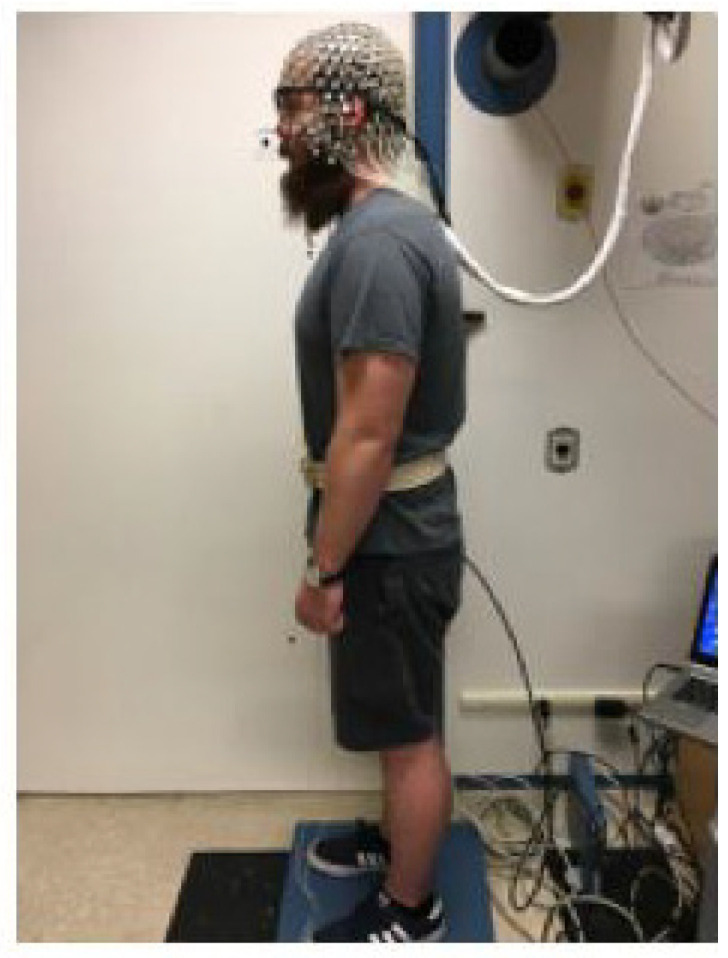
Experimental protocol for standing with eyes open. The person pictured provided informed consent for publication of the image.

**Figure 2 sensors-22-05594-f002:**
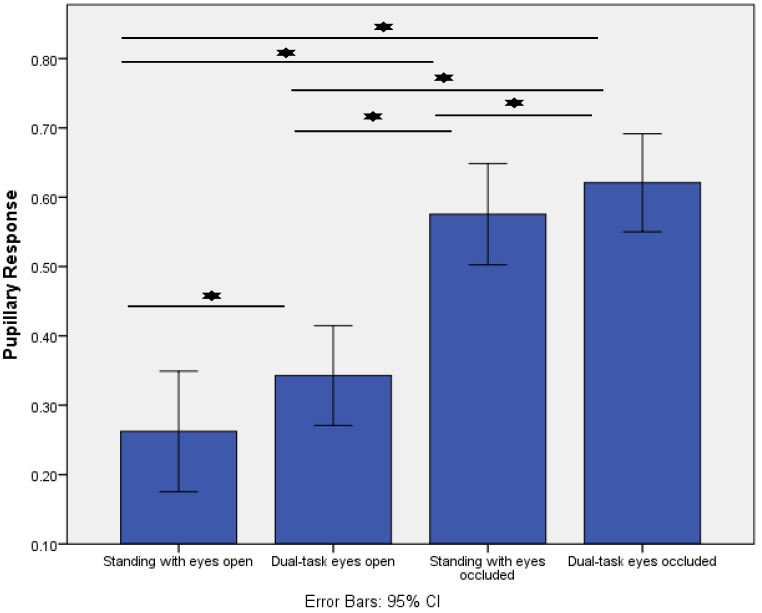
Comparisons of the pupillary response (Index of Cognitive Activity, scale range 0–1) across the four different standing conditions. * indicates *p* < 0.01.

**Figure 3 sensors-22-05594-f003:**
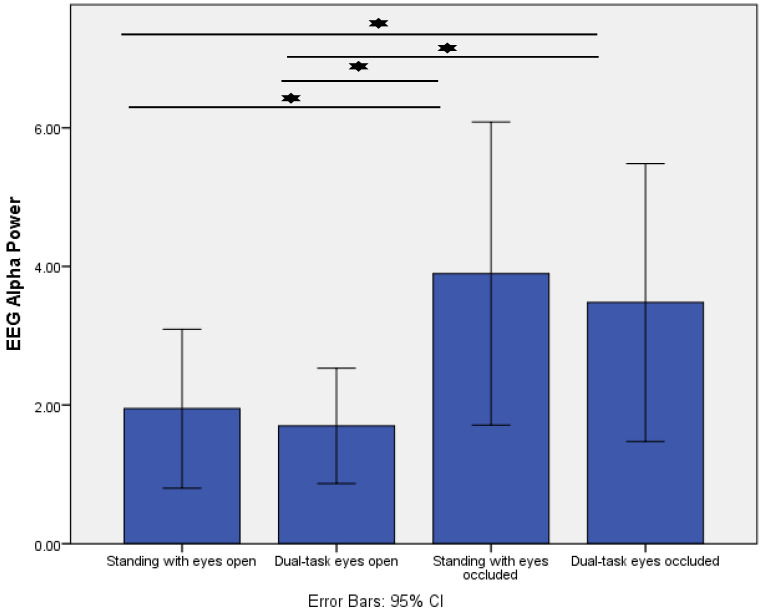
Comparison of posterior EEG alpha power (µV^2^/Hz) across the four different standing conditions. * indicates *p* < 0.01.

**Figure 4 sensors-22-05594-f004:**
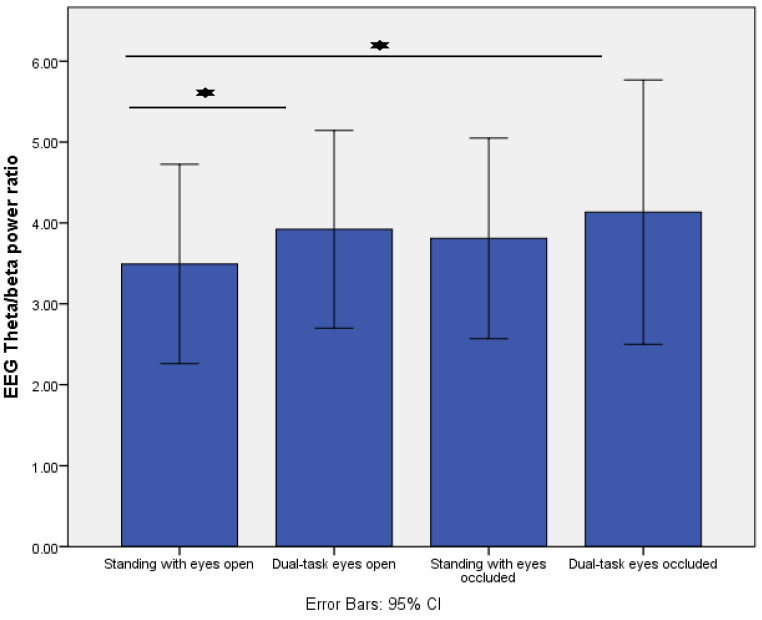
Comparison of fronto-central EEG theta/beta power ratio across the four different standing conditions. * indicates *p* < 0.01.

**Figure 5 sensors-22-05594-f005:**
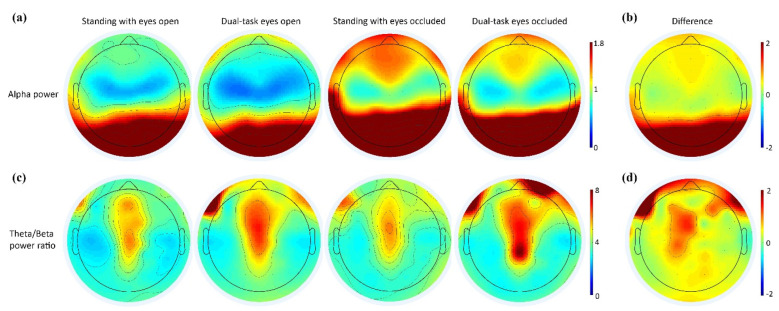
Scalp maps of alpha power (µV^2^/Hz) and theta/beta power ratio distribution across the channels. Warmer or cooler colors indicate more or less power in each frequency band, respectively. (**a**) The topographical distribution of the absolute spectral power (µV^2^/Hz) of alpha demonstrated a significant increase in the posterior regions with eyes-occluded conditions. (**b**) The difference in the alpha power from standing with eyes open to standing with eyes occluded. (**c**) The topographical distribution of the absolute spectral power of the theta/beta power ratio demonstrated a significant increase in the fronto-central regions with dual-tasking conditions compared to standing conditions. (**d**) The difference in the theta/beta ratio from standing with eyes open to dual-task eyes open.

**Figure 6 sensors-22-05594-f006:**
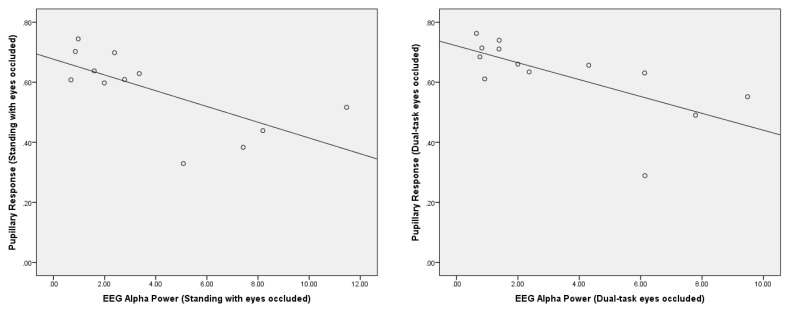
Correlation analysis between the pupillary response and EEG posterior alpha power (µV^2^/Hz) during standing with eyes occluded (*r* = −0.69, *p* = 0.01) and dual-task eyes occluded (*r* = −0.67, *p* = 0.01).

**Table 1 sensors-22-05594-t001:** Demographic characteristics.

Variables	Healthy Young Adults (*n* = 15)
Age (years)	25.6 ± 2.9
Education (years)	18.6 ± 1.6
Sex (female/male)	5/10
Montreal Cognitive Assessment, out of 30	28.8 ± 1.5
Handedness (right/left)	15/0

The results are presented as mean ± standard deviation, except for the sex and handedness variables.

## Data Availability

The datasets generated during and/or analyzed during the current study are available from the corresponding author on reasonable request.

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
