# Peer review of "Cortical Correlates of Increased Postural Task Difficulty in Young Adults: A Combined Pupillometry and EEG Study"

_sensors, 2022, doi:10.3390/s22155594_

Round 1

Reviewer 1 Report

The authors designed an interesting study with a possible high clinical applicability for the future.

Introduction is well written with data from previous studies and accompanied with an extensive bibliographical note. The aim of the study is clearly described.

Methods are systematically described. Inclusion and exclusion criteria are clear. Protocol of the study is extensively described.

Results are clearly stated, following the aim of the study. The results are accompanied by figures relevant for the obtained statistical data.

The article has an extensive and actual bibliography.

Reviewer 2 Report

That is an interesting paper.

1. I suggest to work out more clearly the purpose of the study and the main results: Could you provide in bullet form the main findings of your study?

2. For the reader less familiar with the pupillary response: could you provide images of the main pupillary responses in the same individual, how the measurement occurred and correlate it to the EEG?

3. What was the duration of each EEG measurement? How did you account for sleepiness, as it causes EEG slowing.
